# The Relationship between the Family Functioning of Individuals with Drug Addiction and Relapse Tendency: A Moderated Mediation Model

**DOI:** 10.3390/ijerph18020625

**Published:** 2021-01-13

**Authors:** Xiaoqing Zeng, Chuyi Tan

**Affiliations:** Institute of Psychology, School of Psychology, Jiangxi Normal University, Nanchang 330022, China; tanchuyi2019@163.com

**Keywords:** individuals with drug addiction, family functioning, relapse tendency, psychological capital, life history strategy

## Abstract

To explore the relationship between family functioning, psychological capital, life history strategy, and relapse tendency of individuals with drug addiction, 842 individuals with drug addiction completed a questionnaire. The results showed that (1) there was a significant negative correlation between the family functioning of individuals with drug addiction and their relapse tendency; (2) psychological capital played an intermediary role between family functioning and relapse tendency; and (3) life history strategy regulated the mediating effect of psychological capital. The results of this study suggest that family members should collaborate with drug addiction treatment centers and participate in the education and treatment process to help reduce drug relapse tendency. Increasing the psychological capital and self-efficacy of individuals with drug addiction through group psychological counseling and psychological education courses could also reduce drug relapse tendency.

## 1. Introduction

At the end of 2018, there were 2.404 million individuals with drug addiction in China, and 504,000 individuals had relapsed [1]. Drug relapse refers to the physical and psychological use, intake, or abuse of psychoactive substances by individuals with drug addiction after drug addiction treatment and rehabilitation [2]. Currently, the theoretical research on relapse is not complete or sufficiently mature, and the influencing factors and neural mechanisms that lead to addiction and relapse are unclear [3]. Compulsory isolation detoxification is currently the most important detoxification method in China. This approach is the key to reducing the relapse tendency of individuals with drug addiction after education and correction during the detoxification period [4,5]. Therefore, relapse tendency is an important index to measure the effect of education correction. Relapse tendency refers to the possibility of and to engage in relapse behavior. The higher the relapse tendency is, the higher the probability of relapse. For an individual with drug addiction who is still in the detoxification stage, relapse tendency can be used as one of the key indicators to predict relapse behavior, and thus, relapse tendency has important practical value for evaluating the effects of education and correction. The purpose of this study was to investigate the psychological pathways that influence relapse tendency.

### 1.1. Family Functioning and Relapse Tendency

The family environment is of great importance to the individual’s psychological behavior. Family functioning refers to a model in which family members can obtain the necessary material and spiritual conditions from their family to advance and promote their physical, mental and social development in a healthy and beneficial direction [6]. Family functioning is closely related to individual drug use. Studies have shown that individuals are less likely to have problem behaviors when they interact well with family members, especially their parents [7]. Family functioning plays an important role in male drug abuse [8]. Family functioning has also been found to be related to an individual’s illegal behavior and substance use behavior [9,10]. Compared with adolescents with poor family functioning, adolescents with good family functioning have a lower possibility and risk of drug use [11,12,13,14]. Therefore, this study proposes Hypothesis 1 as follows: Family functioning can negatively predict relapse tendency. However, the specific mechanism of action in the relationship between family functioning and relapse tendency is still unclear, and an analysis of mediating and moderating effects can more clearly reveal how independent variables affect dependent variables [15].

### 1.2. The Mediating Role of Psychological Capital

Psychological capital refers to a kind of positive resource or healthy internal psychological state in the process of an individual’s development, progress and growth [16]. Psychological capital consists of four factors: hope (individuals can adhere to their goals and change their approach to achieving them when necessary), resilience (when encountering problems or setbacks, individuals can persist and maintain their efforts to achieve success), optimism (individuals make positive attributions to present and future success), and self-efficacy (individuals have confidence in certain efforts to complete challenging tasks) [17].

Previous studies have shown that family functioning has a significant positive predictive effect on optimism [18]. Family functioning can significantly positively predict an individual’s levels of hope [19,20], resilience [13,21] and self-efficacy [22]. According to family system theory, the health of the functioning of the family system has an important impact on individuals [23]. The higher the health of the family system functioning, the healthier its members will be in terms of both their physical and mental states [23]. Therefore, parents with a positive sense of hope are more likely to cultivate positive, hopeful offspring [24], which indicates that the level of psychological capital of their offspring may also be higher.

Many studies have shown that various factors of psychological capital are closely related to substance abuse. Self-efficacy significantly negatively predicts the relapse tendency of individuals with drug addiction [25,26]. Optimism and hope are important decisive factors for avoiding substance use [27]. Individuals with a higher level of hope have a stronger sense of self-efficacy and a stronger motivation for withdrawal [28]. In addition, resilience is a protective factor of drug use [29,30], and it significantly negatively predicts the risk of relapse of individuals with drug addiction [31]. In short, positive psychological resources, such as psychological capital, are effective protective factors against substance abuse [32]. Therefore, this study proposes Hypothesis 2 as follows: Psychological capital plays a mediating role between family functioning and relapse tendency.

### 1.3. The Moderating Effect of Life History Strategy

From the perspective of evolutionary psychology, life history theory provides a theoretical framework for individuals’ resource allocation differences in mating strategy, risk-taking behaviors, reproductive development, health and other related behaviors and their outcomes [33]. After these aspects are concretized, they become indicators of differences in life history strategy among individuals. Life history strategies fall on a continuum from fast to slow speed. An individual may adopt a fast life history strategy or a slow life history strategy [34]. Individuals who adopt fast life history strategies are more impulsive and tend to commit more social violations. They are less future oriented, pay more attention to finding a spouse and producing more offspring, invest less time and energy in their offspring and exhibit sexual promiscuity and risk-taking behavior. Individuals who adopt slow life history strategies are less impulsive and generally abide by the law [33,35]. To better adapt to their living environment, individuals often choose different resource allocation models; that is, they adopt different life history strategies.

Although family functioning may influence the relapse tendency of individuals with drug addiction through psychological capital, the influence may vary across individuals with different life history strategies. According to the protective factor model [36], in the prediction of outcome variables, there may be interactions among different protective factors; that is, the predictive effect of one protective factor (such as family functioning) on outcome variables (such as psychological capital) may vary with the level of another protective factor (such as life history strategy). However, there are two different modes of interaction. First, according to the “promotion hypothesis”, the predictive effect of one protective factor on the outcome variable may be enhanced by the other [37,38]. From the perspective of this hypothesis, compared with individuals with poor family functioning, individuals who have good family functioning and who adopt slow life history strategies have higher psychological capital. Second, according to the “exclusion hypothesis”, which contrasts with the “promotion hypothesis”, the predictive effect of one protective factor on the outcome variable may be weakened by another protective factor [37,38]. From the perspective of this hypothesis, compared with individuals with poor family functioning, individuals who have good family functioning and who adopt slow life history strategies have lower psychological capital, and individuals who have good family functioning and who adopt fast life history strategies have higher psychological capital. Therefore, this study proposes Hypothesis 3 as follows: The mediating role of psychological capital between family functioning and relapse tendency is regulated by life history strategy.

In conclusion, this study aims to construct a moderated mediation model (as shown in Figure 1) to further explore the influence of family functioning on relapse tendency and its internal mechanism.

## 2. Method

### 2.1. Participants

The questionnaire was distributed to 900 individuals with drug addiction in three compulsory isolation centers. A total of 842 valid questionnaires were collected, and the effective recovery rate was 93.56%. There were 594 males and 248 females with an average age of 34.56 years (SD = 8.29). Table 1 shows the demographic characteristics of the participants. The Ethics Committee of the School of Psychology at Jiangxi Normal University approved this study.

### 2.2. Research Tools

#### 2.2.1. Family Functioning Scale

The “general functioning” subscale of the Chinese version of the Family Assessment Device [39,40] was used to evaluate the health of the participants’ family functioning. The scale includes 12 items: 6 items are used to evaluate healthy family functioning, and 6 items are used to evaluate unhealthy family functioning. The items are scored on a 4-point scale, where 1 is “not at all like my home” and 4 is “very much like my home”. Negatively worded items were reverse scored. The total score of all items was calculated to obtain the total family functioning score. The higher the total score is, the better the family functioning. In this study, the scale had good internal consistency (Cronbach’s α = 0.796).

#### 2.2.2. Relapse Tendency Questionnaire

The Relapse Tendency Questionnaire has 18 items in five dimensions: confidence in drug treatment, actual influence of drugs, objective environment, degree of physical and mental damage and support system [41]. The questionnaire is scored on a 6-point scale (where 0 is the lowest degree and 5 is the highest degree). The higher the total score is, the higher the relapse tendency. In this study, the questionnaire had good internal consistency (Cronbach’s α = 0.906).

#### 2.2.3. Psychological Capital Questionnaire

The Chinese Psychological Capital Questionnaire (PPQ) has 26 items in 4 dimensions: self-efficacy, resilience, hope, and optimism [42]. A 7-point scale is used (where 1 is “completely nonconforming” and 7 is “completely conforming”). The average score of all items was calculated after negatively worded questions were reverse scored. The higher the score is, the higher the individual’s psychological capital. In this study, Cronbach’sαcoefficients of “self-efficacy”, “resilience”, “hope” and “optimism” were 0.82, 0.60, 0.65 and 0.78, respectively, and the questionnaire had good internal consistency (Cronbach’s α = 0.883).

#### 2.2.4. Life History Strategy Scale

The Mini-K Scale, which is a subscale of the Arizona Life History Scale, was used [43]. There are 20 items in this scale that are scored on a 7-point scale from −3 (“strongly disagree”) to 3 (“strongly agree”). A positive total score indicates a slow life history strategy, and a negative total indicates a fast life history strategy. In this study, the scale had good internal consistency (Cronbach’s α = 0.894).

#### 2.2.5. Control Variables

The differences in the demographic variables and family functioning, relapse propensity, psychological capital and life history strategies were tested, and it was found that there were significant differences in both life history strategies and relapse propensity by gender (*t* = 7.49, *p* < 0.001, *d* = 0.55; *t* = 4.59, *p* < 0.001, *d* = 0.32) and a significant difference in psychological capital by age (*F*_(42,799)_ = 1.63, *p* < 0.01, *η*^2^ = 0.08). In addition, family functioning, psychological capital and life history strategies significantly differed by education level (*F*_(5,836)_ = 2.36, *p* < 0.05, *η*^2^ = 0.01; *F*_(5,836)_ = 3.16, *p* < 0.01, *η*^2^ = 0.02; *F*_(5,836)_ = 2.44, *p* < 0.05, *η*^2^ = 0.01). Therefore, this study used demographic variables such as gender, age, and educational level as control variables.

### 2.3. Data Analysis

First, Excel 2010 was used to organize the data, and then the statistical software SPSS 21.0 and Amos 24.0 were used to analyze the data.

## 3. Results

### 3.1. Control and Inspection of Common Method Variance

In this study, two methods were used to reduce and test the common method variance. First, we adopted three strategies to control the common method variance: anonymous completion, partial reverse scoring, and different scoring methods. Second, two statistical methods were used to test the common method variance: (1) the Harman single-factor test, which showed that there are 29 factors with characteristic roots greater than 1, accounting for 74.839% of the variance, and that the first factor explains 14.678% of variance, which is less than 40% [44]; (2) confirmatory factor analysis, which was used to load all measurement items load on the same latent factor and which showed that the model fit is not good (χ^2^ = 24,987.45, *df* = 2849, χ^2^/*df* = 8.77, *p* < 0.001; RMSEA = 0.10; CFI = 0.12; TLI = 0.12; SRMR = 0.17), indicating that all measurements are different and belong to one factor. Therefore, there is no serious common method variance in this study.

### 3.2. Descriptive Statistics and Correlation Analysis

The average, standard deviation, and correlation coefficient of each variable are shown in Table 2. Family functioning is significantly positively correlated with psychological capital and life history strategy and negatively correlated with relapse tendency. Relapse tendency is negatively correlated with psychological capital and life history strategy. Psychological capital is significantly positively correlated with life history strategy.

### 3.3. Moderated Mediation Model Test

This study used the SPSS macro program process [45] to test the mediating role of psychological capital between family functioning and relapse tendency and to determine whether the first half of the path of this mediating role is regulated by life history strategy.

The results are shown in Table 3. Family functioning significantly negatively predicts relapse tendency (*β* = −0.12, *p* < 0.001) and significantly negatively predicts psychological capital (*β* = 0.15, *p* < 0.001), and psychological capital has a significant negative predictive effect on relapse tendency (*β* = −0.23, *p* < 0.001). This result shows that psychological capital mediates the influence of family functioning on the relapse tendency of individuals with drug addiction. Thus, Hypothesis 1 is verified. In addition, the interaction between family functioning and life history strategy has a significant predictive effect on psychological capital (*β* = −0.12, *p* < 0.001), indicating that the influence of family functioning on psychological capital is regulated by life history strategy. Therefore, Hypothesis 2 was verified.

To better explain the mediation model with regulation, life history strategy was divided into three levels, namely slow life history strategy, fast life history strategy, and medium-speed life history strategy, which were defined as one standard deviation above, one standard deviation below, and one standard deviation between the average, respectively. The value of the mediation effect of psychological capital and the 95% bootstrap confidence interval are shown in Table 4. A simple slopes analysis was used to investigate the role of life history strategy in the relationship between family functioning and psychological capital. The specific regulatory effect is shown in Figure 2. Family functioning has a significant effect on psychological capital (*β* = 0.27, *t* = 6.00, *p* < 0.001, 95% Bootstrap CI = [0.18, 0.36]) when individuals with drug addiction adopt fast life history strategies, while family functioning has no significant effect on psychological capital (*β* = 0.03, *t* = 0.61, *p* > 0.05, 95% Bootstrap CI = [−0.06, 0.12]). Therefore, family functioning has a greater impact on the psychological capital of individuals with drug addiction who adopt fast life history strategies than on individuals with drug addiction who adopt slow life history strategies.

## 4. Discussion

### 4.1. The Mediating Role of Psychological Capital in the Relationship between Family Functioning and Relapse Tendency of Individuals with Drug Addiction

This study found that family functioning has a significant negative predictive effect on the relapse tendency of individuals with drug addiction, which is consistent with previous research results [46,47]. The results of this study support McMaster’s family functioning model [48]. Family members’ role allocation, mutual communication, ability to solve practical problems and other aspects can show whether the individual’s family functioning is good [6]. Families with good family functioning should have a warm emotional atmosphere, open communication among family members, clear role allocation and effective problem solving [48]. Individuals with good family functioning are more inclined to abide by social rules and norms and are less likely to commit crimes and other antisocial behaviors [49]. The better the functioning of the family, the higher the cohesion and harmony of the family; thus, the mental health of the family members will be higher, and there will be fewer problem behaviors [6,50]. In addition, family system theory [51,52] suggests that the family has the function of maintaining the stability of the family system. When individuals feel that there is conflict between their parents and their marriage, i.e., that the family is not functioning properly, they are more likely to exhibit problem behaviors [53]. In this study, the education level of the subjects was concentrated in the compulsory education stage, which indicated that the subjects left school and their families prematurely and entered society. Some studies have shown that risk factors for substance abuse include premature separation from the family and a lack of parental supervision [54]. In addition, in adolescence, individuals begin to spend more time with their peers, and peers become more important external factors affecting individuals’ behaviors. To integrate into their peer group, seek the recognition and support of the peer group, and prevent discrimination and exclusion from their peers, adolescents usually make their behaviors (such as smoking) consistent with those of their peers [55]. In this study, the average age of the subjects when they smoked for the first time was 17 years old, which is an important transition stage from adolescence to adulthood. With the rapid development of their bodies and minds, adolescents have a growing sense of entering adulthood. They are eager to understand unknown aspects of the external world. They easily become curious about new things. Exploring and trying new things can help them obtain greater happiness. Drugs easily trigger the “forbidden fruit effect” for young people, and young people are increasingly curious and motivated to approach them. In addition, young people’s abilities to resist harmful temptations from the outside world are not strong enough, which makes them more likely to take drugs. Studies have shown that negative peer interaction can significantly predict individual drug use behavior [56,57,58]. This study and previous studies show that family functioning can negatively predict the relapse tendency of individuals with drug addiction, which is also indicative of whether family functioning is healthy and whether the family can provide effective family social support. Effective family support can play a buffering role, help individuals with drug addiction obtain a sense of support, reshape their beliefs and confidence, and reduce their relapse tendency.

Second, this study also reveals that psychological capital plays a partial mediating role between family functioning and relapse tendency. On the one hand, family functioning can positively predict the psychological capital of individuals with drug addiction, which is consistent with previous studies [18,19,21,22]. Optimism and self-efficacy are two core factors of an individual’s sense of mastery. Individuals with family dysfunction have a lower sense of mastery; that is, their optimism and self-efficacy are lower [59]. Good family functioning means that there is close emotional connection and communication between parents and children in the family, which is conducive to the healthy development of individuals’ mental health, which in turn, increases psychological capital. The ecosystem model [60] holds that individuals’ development is affected by four different ecological spatial environments: microsystems, mesosystems, appearance systems, and macrosystems. The microsystem directly acts on individuals; examples are the family atmosphere and school environment. The mesosystem is the interaction and influence between more than two microscopic systems. The appearance system indirectly acts on individuals. Finally, the macrosystem refers to culture, which has a special influence on individuals through the social context. The characteristics of individuals, such as psychological function and coping style, interact with these systems and jointly affect individuals’ development and behaviors. The family environment is a microsystem that directly affects individuals’ behaviors and development. Therefore, the family functioning of individuals with drug addiction negatively predicts their relapse tendency at the micro level. On the other hand, psychological capital can negatively predict the relapse tendency of individuals with drug addiction, which is consistent with previous studies [32,61,62]. According to social cognitive theory, the internal psychological factors of individuals, the external environment and individuals’ behaviors interact with each other [63]. Psychological capital is an individual’s positive evaluation of the environment and positive expectation of the possibility of success. Psychological capital is based on persistence and efforts to achieve goals [64]; it contributes to the level of the individual’s internal psychological state, and it is an inherent psychological characteristic of the individual. The interaction between psychological capital and the external environment, i.e., family functioning, will affect the relapse tendency of individuals with drug addiction. When individuals with drug addiction have high psychological capital, they are full of optimism and hope for the future, maintain a positive attitude, have positive expectations and evaluations of life, have a high sense of self-efficacy, have strong ideas and motivation for drug treatment, and can resist the temptation of drugs; when they encounter setbacks, their higher resilience can help them to continue to adhere to and strive for the goal of detoxification. Hence, their relapse tendency will be reduced. Therefore, the relapse tendency of individuals with drug addiction is not only negatively predicted by their family functioning but also indirectly predicted by their psychological capital.

### 4.2. The Moderating Effect of Life History Strategy

This study further found that life history strategy regulates the relationship between family functioning and psychological capital. Specifically, compared with individuals with drug addiction who adopt slow life history strategies, the family functioning of individuals with drug addiction who adopt fast life history strategies has a greater significant impact on their psychological capital. Therefore, this study supports the “exclusion hypothesis” rather than the “promotion hypothesis” in the protective factor model. Some studies have shown that life history strategy can significantly positively predict the level of an individual’s mental health [65] and that psychological capital significantly positively predicts mental health [42], indicating that both life history strategy and psychological capital are related to mental health level, which may mean that an individual’s life history strategy is related to his or her psychological capital level. According to life history theory, attachment between parents and children is closely related to the life history strategies adopted by individuals [66]. Individuals with secure attachment to their parents usually adopt slow life history strategies, whereas those who form insecure attachment with their parents usually adopt fast life history strategies [66]. Good family functioning can positively predict parent-child attachment [67]. Good parent-child attachment can improve an individual’s optimism, hope, self-efficacy and resilience [68,69,70,71,72], thus increasing psychological capital. Therefore, individuals who adopt fast life history strategies may have fewer resources, such as time and energy invested by their parents, resulting in insecure attachment and lower psychological capital. When individuals have good family functioning, their psychological capital will be more strongly promoted; however, those who adopt slow life history strategies often have parents who invested more resources to form a secure form of attachment, and they show higher psychological capital and a weaker effect of family functioning on the improvement of their psychological capital. In this study, 29.8% of the subjects reported that their parents had poor marital status (including divorced, remarried and widowed), and 84.1% of them reported that they were children when their parents divorced. The occurrence of problems in one’s parents’ marriage is a stressful event in an individual’s life that will directly affect his or her mental health [73], resulting in low psychological capital.

### 4.3. Implications and Limitations

The results of this study suggest that to reduce the relapse tendency of individuals with drug addiction, we should first educate and guide the family members of individuals with drug addiction to care more about them, provide them with the necessary psychological support and encouragement, create a warm and comfortable family atmosphere for them, and thereby improve their family functioning. Second, the mental health level of individuals with drug addiction can be improved by means of group psychological counseling with a focus on the psychological capital.

There are some limitations in this study. First, the subjects of the study were individuals with drug addiction in mainland China, and the results may not be applicable to the individuals with drug addiction in other countries and regions. Second, this study involved horizontal research. A future study could employ longitudinal research and select some subjects for follow-up investigation. Third, this study found that psychological capital plays a partial mediating role between family functioning and relapse tendency, and there may be other mediating variables between the two. The influence of multiple mediating variables on relapse tendency of individuals with drug addiction needs to be explored.

## 5. Conclusions

Based on our analysis and discussion, the current study suggests that the family functioning of individuals with drug addiction directly affects the relapse tendency, while psychological capital has an indirect effect. After gender, age, and educational level are controlled, the results indicate that psychological capital plays a partial mediating role between family functioning and relapse tendency. According to the results of this study, the role of psychological capital between family functioning and relapse tendency is moderated by life history strategy.

## Figures and Tables

**Figure 1 ijerph-18-00625-f001:**
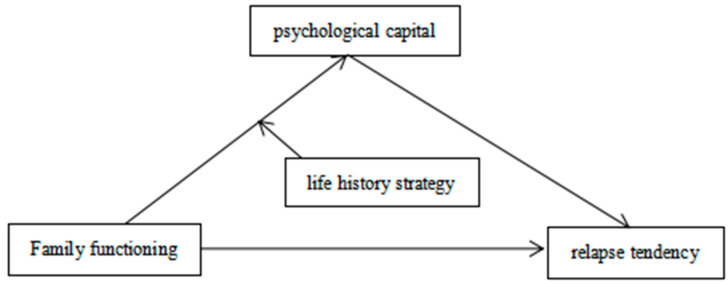
A hypothetical model of the effect of family functioning on relapse tendency.

**Figure 2 ijerph-18-00625-f002:**
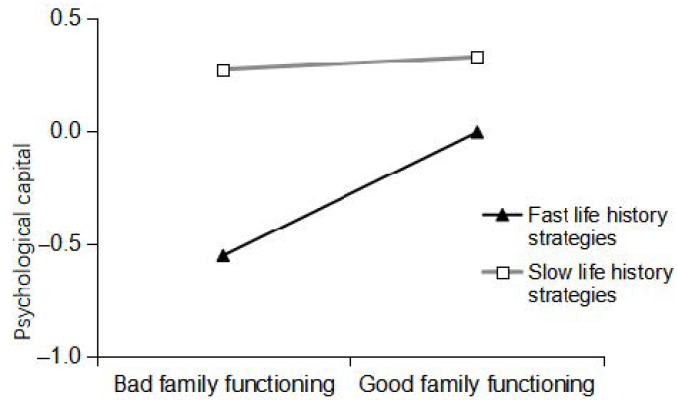
Diagram of the moderating effect of life history strategies on the relationship between family functioning and psychological capital.

**Table 1 ijerph-18-00625-t001:** Demographic characteristic of the participants (n = 842).

Variables	N	%
**Gender**		
Male	594	70.5
Female	248	29.5
**Education level**		
Primary school or below	144	17.1
Junior high school	501	59.5
High school or secondary school	163	19.4
College	24	2.9
Bachelor’s degree or above	10	1.2
**Type of drug use**		
Methamphetamine	607	72.1
K powder	140	16.6
Heroin	79	9.4
other drugs	16	1.9
**Parents’ marital status**		
married	592	70.3
divorced	120	14.3
remarried	31	3.7
widowed	99	11.8
**parental relationships**		
very good	353	41.9
Good	228	27.1
moderate	180	21.4
poor	54	6.4
Very poor	27	3.2

**Table 2 ijerph-18-00625-t002:** Means, standard deviations, and correlation matrix of all variables.

	*M*	*SD*	1	2	3	4
1. Family Functioning	3.42	0.65	1			
2. Relapse Tendency	1.46	0.77	−0.15 ***	1		
3. Psychological Capital	0.17	0.03	0.18 ***	−0.25 ***	1	
4. Life History Strategy	0.20	0.97	0.08 *	−0.01	0.27 ***	1

*N**=* 842, * *p* < 0.05, *** *p* < 0.001, the same below.

**Table 3 ijerph-18-00625-t003:** Regression Analysis of the mediating role of life history strategies in mediating family functioning and relapse tendency.

Regression Equation		Overall Model Fit			Regression Coefficient Significance			
Outcome	Predictor	*R*	*R^2^*	*F*	*β*	LLCI	ULCI	*t*
psychological capital	Gender	0.34	0.12	18.53 ***	0.10	−0.05	0.25	1.36
	Age				0.00	−0.01	0.01	0.56
	Education level				0.10	0.01	0.18	2.28 *
	family functioning				0.15	0.09	0.22	4.61 ***
	life history strategy				0.29	0.22	0.36	8.48 ***
	family functioning × life history strategy				−0.12	−0.19	−0.06	−3.68 ***
relapse tendency	Gender	0.31	0.10	17.98 ***	−0.34	−0.48	−0.19	−4.61 ***
	Age				0.00	−0.01	0.01	0.21
	Education level				0.00	−0.09	0.09	0.03
	psychological capital				−0.23	−0.30	−0.17	−6.89 ***
	family functioning				−0.12	−0.18	−0.05	−3.46 ***

All variables in the model were brought into the regression equation after standardized treatment, the same below. Gender: female = 0, male = 1.

**Table 4 ijerph-18-00625-t004:** The mediating effect of psychological capital under different life history strategy levels.

Life History Strategy Level	Effect Size	Bootstrap Standard Error	*t*	LLCI	ULCI
*M* − 1*SD*	0.27	0.05	6.00 ***	0.18	0.36
*M* ± 1*SD*	0.15	0.03	4.61 ***	0.09	0.22
*M* + 1*SD*	0.03	0.05	0.61	−0.06	0.12

## Data Availability

The data presented in this study are available on request from the corresponding author. The data are not publicly available due to three reasons. First, the research data involves the privacy of individuals with drug addiction. Second, the informed consent form stipulates that the data will not be disclosed to any third party. Last, the publication of the research data needs the consent and permission of the rehabilitation institutions.

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
