# Peer review of "The Relationship between the Family Functioning of Individuals with Drug Addiction and Relapse Tendency: A Moderated Mediation Model"

_ijerph, 2021, doi:10.3390/ijerph18020625_

Round 1
Reviewer 1 Report
This paper explored how family functioning, psychological capital, and life history strategy affect relapse tendency of individuals. I have some suggestions on the writing & structure of this paper:
- some of the content in Introduction was repeated in discussion. Please remove the redundant sentences.
- The introduction is too lengthy - the authors can incorporate some of the introduction parts to results session.
- This study lacks novelty - please describe how this study differs from previous ones.
Author Response
Thank you for your comments concerning our manuscript entitled “The relationship between the family functioning of individuals with drug addiction and relapse tendency: a moderated mediation model” ( ID: ijerph-1036577).Those comments are valuable and very helpful for revising and improving our paper, as well as the important guiding significance to our researches. We have studied comments carefully and have made correction which we hope meet with approval. Revised portion are marked in red in the paper.
The main corrections in the paper and responds to reviewer's comments are as following:
- Response to comments:“Remove the redundant sentences.”
Response: It’s really true as Reviewer suggested that some of the content in Introduction was repeated in discussion. Therefore, we deleted some repeated sentences in the discussion.
①Line 236-237, “Many studies have shown that family functioning is the key factor influencing the relapse of individuals with drug addiction after withdrawal.” was deleted.
②Line 238-240, “According to this model, the family can provide the necessary conditions for the healthy development of the body and mind and the socialization of its members, and family functioning has an important impact on the behavior of family members. ” was deleted.
③Line 277, “and that parent-child interaction is more frequent” was deleted.
④Line 286-287, “the most important place for individuals’ growth, and it is” was deleted.
- Response to comments:“Incorporate some of the introduction parts to results session.”
Response: First of all, thank you very much for your valuable comments. We have adjusted and deleted part of the introduction according to your comments. Secondly, according to your comments, we have carefully reviewed our paper and the papers recently published in IJERPH. Most of the published papers only present the results of the research itself in the results part, and do not use other studies to analyze them in the results part. Therefore, we do the same according to the requirements of IJERPH.
The specific changes are as follows:
①The [4] and [5] references were added.
②Line 40-41, “is where individual socialization begins and is an important place for the individual’s overall physical and mental development. It” was deleted.
③Line 67-68, “In a family with positive, healthy family functioning, a goal-oriented thinking process and problem-solving orientation are emphasized.” was deleted.
④Line 69-72, “As a very important psychological quality, psychological capital can reflect one’s inner positive state and is a positive psychological resource for individuals, and it can profoundly affect individuals’ cognition and behaviors. More than 75% of drug relapse problems are considered to be related to a lack of social and family support.” was deleted.
⑤The reference “Zhong, W. F., Guo, Y. X. (2018). Effect of Resilience on Drug Relapse Risk: The Mediating Roles of Perceived Stress and Depression. Chinese Journal of Clinical Psychology, 26(6), 1096-1099+1103. doi:10.16128/j.cnki.1005-3611.2018.06.010” was deleted.
The reference “Li, D., Zhang, W., Li, X., Li, N., & Ye, B. (2012). Gratitude and suicidal ideation and suicide attempts among Chinese Adolescents: Direct, mediated, and moderated effects. Journal of Adolescence, 35(1), 55-66. doi:10.1016/j.adolescence.2011.06.005” was deleted.
The reference “Wang, Y. H., Zhang, W., Peng, J. X., Mo, B. R., Xiong, S. (2009). The Relations of Attachment, Self -Concept and Deliberate Self -Harm in College Students. PSYCHOLOGICAL EXPLORATION, 29(5), 56-61. doi:10.3969/j.issn.1003-5184.2009.05.012 ” was deleted.
⑥Line 77-79, “Psychological capital has a significant negative correlation with substance abuse, and psychological capital can negatively predict substance abuse behavior.” was deleted.
⑦Line 87-88, “use their current resources immediately, ” was deleted.
⑧Line 89-90, “In contrast,” was deleted.
⑨Line 91-93, “They are future-oriented; pay more attention to long-term resource allocation; are less sexually promiscuous; allocate resources for themselves, their spouses and future generations; and invest more time and energy in these people accordingly.” was deleted.
- Response to comments:“Describe how this study differs from previous ones.”
Response: We believe that there are some innovations in this study. Previous studies have found that there are many factors that affect the relapse tendency of individuals with drug addiction, but most previous studies only discussed the influence of family function or psychological capital on relapse tendency. As a matter of fact, these influences are seldomly singular; rather, they appear in groups such that they attract and form building blocks with each other. However, the interaction between family factors and personal factors and that interaction’s internal relationship remain unclear. Based on the family function theory, mediation and moderation analyses can be used to determine the relationships between these variables, thus making it a worthy topic for further study. In addition, existing research has mainly focused on drug addicts of voluntary detoxification and community drug treatment in western countries. Thus, it is necessary to perform research the compulsory isolation of individuals with drug addiction from Asian countries. Finally, studying the joint effects of these variables is the key to understanding the basis of drug addiction and developing effective treatment methods as a next step. Even more important is determining the supportive psychological factors for individuals with drug addiction to provide new ideas for helping individuals with drug addiction more successfully remain abstinent in the long term.
Special thanks to you for your good comments!You can view the changed paper in the attachment.
Reviewer 2 Report
The authors have explored the mediating role of psychological capital in the relationship between family functioning and relapse tendency among individuals with drug addiction. They have also tested the moderating role of life history on the relationship between family functioning and psychological capital.
The article is well written and it shows proper rationale. The analyses are well conducted. Sample size is large.
I have several concerns regarding the references of some theories:
- Family System Theory (line 72). It is not clear what original research the authors refer to.
- McMaster´s Family Functioning model (line 254). It is also not clear the original reference.
In order to clarify this, the authors should include the corresponding reference right after the name of the theory.
I also have some concerns regarding the specific versions of the instruments that the authors used with their sample.
- The psychological capital questionnaire. The authors do not specify what version of the scale was used, or the language of the version.
- Mini-K Scale. The authors do not specify what version they used, or the language of the version.
I also have some concerns regarding the format:
-The authors have not addapted the paper to IJERPH references format.
Conclusions: the authors should draw up the conclusions instead of listing them.
Author Response
Thank you for your comments concerning our manuscript entitled “The relationship between the family functioning of individuals with drug addiction and relapse tendency: a moderated mediation model” ( ID: ijerph-1036577).Those comments are valuable and very helpful for revising and improving our paper, as well as the important guiding significance to our researches. We have studied comments carefully and have made correction which we hope meet with approval. Revised portion are marked in red in the paper.
The main corrections in the paper and responds to reviewer's comments are as following:
- Response to comments:“The references of some theories”
Response: We have added the references of the two theories according to the Reviewer’s comments. About Family System Theory (line 64-65), “[23]” was added. About McMaster´s Family Functioning model(line 226), “[48]” was added.
- Response to comments:“The specific versions of the instruments”
Response: We are very sorry for our negligence of the specific versions of the instruments.
The Psychological Capital Questionnaire we used is a Chinese questionnaire compiled by Chinese scholars Zhang, Zhang and Dong(2010), with 26 questions in total.
Mini-K scale is a Chinese version of 20 questions, which was translated into Chinese by a psychology professor and four graduate students majoring in psychology according to the usage habits of Chinese.
- Response to comments:“Not adapt the paper to IJERPH references format”
Response: We are very sorry for our incorrect references format. We have used Arabic numerals for all references, and the citation format is ascending according to IJERPH references format.
- Response to comments:“Draw up the conclusions instead of listing them”.
Response: We have re-written conclusions part according to Reviewer’s suggestion. The original conclusions have been deleted. “Based on our analysis and discussion, the current study suggests that the family functioning of individuals with drug addiction affects directly the relapse tendency as well as indirectly through psychological capital. After controlling for gender, age and educational level, psychological capital plays a partial mediating role between family functioning and relapse tendency. According to the results of this study, the role of psychological capital between family functioning and relapse tendency is moderated by life history strategy.” was added.
As for the English language and form of this paper, we have carefully checked before submission, and the paper has also been professionally polished by American Journal experts.
Special thanks to you for your good comments! You can view the changed paper in the attachment.